# Microbiological Survey and Evaluation of Antimicrobial Susceptibility Patterns of Microorganisms Obtained from Suspect Cases of Canine Otitis Externa in Gran Canaria, Spain

**DOI:** 10.3390/ani14050742

**Published:** 2024-02-27

**Authors:** Rubén S. Rosales, Ana S. Ramírez, Eduardo Moya-Gil, Sara N. de la Fuente, Alejandro Suárez-Pérez, José B. Poveda

**Affiliations:** 1Instituto Universitario de Sanidad Animal y Seguridad Alimentaria (IUSA), Veterinary Faculty, University of Las Palmas de Gran Canaria, Trasmontaña s/n, 35416 Arucas, Spain; ruben.rosales@ulpgc.es (R.S.R.); jose.poveda@ulpgc.es (J.B.P.); 2Análisis Veterinarios Eurofins, Calle Leopoldo Matos, 18, 35006 Las Palmas de Gran Canaria, Spain; saraniza.hernandezdelafuente@ctes.eurofinseu.com; 3Veterinary Faculty, University of Las Palmas de Gran Canaria, Trasmontaña s/n, 35416 Arucas, Spain; eduardomoya.vet@gmail.com; 4Departamento de Patolología Animal, Producción Animal, Bromatología y Ciencia y Tecnología de los Alimentos, Veterinary Faculty, University of Las Palmas de Gran Canaria, Trasmontaña s/n, 35416 Arucas, Spain; alejandro.suarezperez@ulpgc.es

**Keywords:** antimicrobial resistance, canine otitis externa, multidrug resistance

## Abstract

**Simple Summary:**

Canine otitis externa is a highly frequent disease of dogs that is sometimes painful and, if not properly treated, can progress into chronic cases refractory to antimicrobial or antifungal treatment. In order to apply effective treatments for this pathology, it is essential to understand the current trends in the prevalence and antimicrobial susceptibility of the microorganisms involved. For this reason, a study of dog ear culture clinical samples from 2020 to 2022, obtained from a veterinary laboratory of the island of Gran Canaria, Spain, was performed. Results demonstrated a high prevalence of the most common microorganisms involved in canine otitis. In addition, a high frequency of antimicrobial resistance was observed in the most prevalent bacterial species found (*Staphylococcus pseudintermedius*, *Pseudomonas aeruginosa*, *Proteus mirabilis*, *Escherichia coli*). Resistance to multiple antimicrobial classes in the same bacterial isolate, or multidrug resistance, was also observed. In addition, a high prevalence of *Staphylococcus pseudintermedius* resistant to methicillin was found. This is a concerning finding due to the risk these microorganisms represent to animals and humans. Our results confirm the need for a constant evaluation of pathogens involved in canine otitis externa so effective treatments can be implemented.

**Abstract:**

A retrospective study of microbiological laboratory results from 2020 to 2022, obtained from a veterinary diagnostic laboratory of the island of Gran Canaria, Spain, focused on canine otitis cases, was performed. The objective of this study was to analyze the pathogen distribution, antimicrobial susceptibility, prevalence of multidrug resistant phenotypes and the role of coinfections in otitis cases in order to provide up-to-date evidence that could support effective control strategies for this prevalent pathology. A total of 604 submissions were processed for the diagnosis of canine external otitis. Of the samples analyzed, 472 were positive for bacterial or fungal growth (78.1%; 95% CI: 74.8–81.4%). A total of 558 microbiological diagnoses were obtained, divided in 421 bacterial (75.4%; 95% CI: 71.8–79.0%) and 137 fungal (24.6%; 95% CI: 20.9–28.1%) identifications. *Staphylococcus pseudintermedius*, *Malassezia pachydermatis* and *Pseudomonas aeruginosa* were the most prevalent microorganisms detected in clinical cases of otitis. High level antimicrobial resistance was found for *Pseudomonas aeruginosa* (30.7%), *Proteus mirabilis* (29.4%), *Staphylococcus pseudintermedius* (25.1%) and *Escherichia coli* (19%). Multidrug-resistant phenotypes were observed in 47% of the bacteria isolated. In addition, a 26.4% prevalence of methicillin-resistant *Staphylococcus pseudintermedius* was detected. The high prevalence of antimicrobial resistant phenotypes in these bacteria highlights the current necessity for constant up-to-date prevalence and antimicrobial susceptibility data that can support evidence-based strategies to effectively tackle this animal and public health concern.

## 1. Introduction

Otitis externa is one of the most prevalent pathologies of the ear of dogs worldwide, with prevalence rates ranging from 5 to 20% [1,2]. This disorder is described in the majority of cases as a multifactorial disease, normally associated with a primary damage or alteration of the ear environment caused by various entities, such as foreign bodies, endocrinopathies, autoimmune disease, parasites or allergies [3], that, in combination with predisposing factors such as breed and ear conformation, facilitate colonization by opportunistic pathogens [4,5].

Canine otitis externa has relevant welfare implications, as it is frequently associated with discomfort and pain. In addition, if untreated or unproperly managed, it can lead to chronic changes that will enable the presence of more frequent ear infections, which can increase in severity and become refractory to classic drug-based treatments [6].

For this reason, many research efforts have been focused on understanding the role of the commensal microbiota of the ear of dogs in the development of otitis externa. Culture-based identification, and more recently, next-generation sequencing-based microbiome analysis, have identified multiple Gram-positive and Gram-negative bacterial species linked to otitis externa, including *Staphylococcus* spp., *Pseudomonas* spp., *Proteus* spp., *Escherichia coli* and *Corynebacterium* spp., in addition to opportunistic commensal yeasts, such as *Malassezia pachydermatis* and *Candida albicans* [3,7,8,9,10,11], that in normal conditions inhabit the ear epithelium without causing any damage.

Besides the potential primary factors linked to the development of canine otitis externa, some of these microorganisms have the ability to express environmental mechanisms of resistance, such as biofilms, that can complicate both the correct action of antimicrobial treatments and the development of protective immunity. For example, the ability of some *Pseudomonas* (*P*.) *aeruginosa* isolates¸ the most prevalent Gram-negative bacteria found in cases of canine otitis [2], to produce biofilms can reach up to 40%, leading to increases in the antimicrobial concentration needed for an effective resolution of the infection [12]. This resistant mechanism has also been described in the two most prevalent pathogens isolated in cases of canine otitis externa, *Staphylococcus* (*S*.) *pseudintermedius* and *M. pachydermatis* [6,13]. Therefore, when treatment protocols for canine otitis externa are not adapted to the role of biofilm-producing microorganisms in the infection, the potential suboptimal use of antimicrobial treatments can facilitate the appearance of multidrug-resistant (MDR) microorganisms, one of the main threats to public health worldwide and a central concern for One Health initiatives [14]. In addition, some studies suggest very low adherence to evidence when prescribing antimicrobials in veterinary practice, in combination with the frequent administration of broad-spectrum drugs (e.g., amoxicillin-clavulanate and first generation cephalosporins) as first option treatments [15]. Both strategies are classic predisposing factors for the development of antimicrobial-resistant microorganisms.

The development of antimicrobial resistance (AMR) in bacteria and the emergence of MDR microorganisms has both animal welfare and public health connotations. Dogs carrying MDR pathogens have a higher risk of being involved in multiple failed treatment regimens [6] and in some cases can only be treated by using last-resort surgical procedures as the only therapeutic option. The public health, or One Health, concern associated with the presence of AMR and MDR pathogens lies in the zoonotic potential of some of the most common bacteria isolated from cases of canine otitis externa. *P. multocida*, *S. pseudintermedius* and *Staphylococcus* (*S*.) *aureus* have been described as the most common zoonotic microorganisms involved in spread between human and pets [16]. *S. aureus* has been usually linked to cases of human-to-companion animal transmission, while *P. multocida* and *S. pseudintermedius* are most frequently found in cases of pet-to-human zoonoses [16]. Both pathogens have been linked to various clinically relevant antimicrobial resistant phenotypes. For example, in the case of *S. pseudintermedius*, methicillin-resistant phenotypes (MRSP) have emerged worldwide in recent years. MRSP population are resistant to all β-lactams, and in most cases also carry other resistance determinants, which usually classifies MRSP also as MDR isolates [17]. Regarding *P. aeruginosa*, the expansion of MDR phenotypes is a common feature, due to a combination of intrinsic and acquired resistance mechanisms, greatly limiting the therapeutic repertoire against this pathogen [1,18].

Due to the changing nature of AMR and the constant emergence of novel pathogens and MDR phenotypes, it is critical to maintain up-to-date knowledge regarding the microorganisms involved in canine otitis externa and antimicrobial susceptibility phenotypes in order to facilitate the use of evidence-based therapies. For this reason, the objective of the present study is to descriptively analyze the pathogen distribution, antimicrobial susceptibility, prevalence of MDR phenotypes and the role of coinfections in a cohort of diagnostic results obtained from a commercial veterinary diagnostic laboratory of the island of Gran Canaria, Spain.

## 2. Materials and Methods

### 2.1. Data Collection

The database analyzed as part of this retrospective study was obtained from a veterinary clinical diagnostics laboratory (Animal Lab S.L.), in Gran Canaria, Canary Islands, Spain, where more than 200.000 dogs are officially registered [19]. All the data were collected from clinical cases of canine otitis externa, routinely submitted to the laboratory for analysis from 2020 to 2022 and recorded in their laboratory information system (Modulab^®^, Werfen, Spain). The extracted variables included year, breed, sex, age, microbiological identification and antimicrobial susceptibility phenotypes. Age was additionally codified in three groups (<1 year; 1 to 5 years, >5 years) as previously described [20]. In addition, the presence of co-infections in positive cultures was also recorded. All data were collected and normalized in Microsoft Excel^®^ prior statistical analysis.

### 2.2. Microbiological Identification and Antimicrobial Susceptibility Testing (AST)

Ear swabs received at the laboratory were routinely plated in Chocolate agar, Cled agar, MacConkey agar and Sabouraud agar (bioMérieux, Marcy l’Etoile, France).

After subculture, bacteria were identified using the Vitek^®^ 2 automated system’s GP and GN cards (bioMérieux, Marcy l’Etoile, France). AST was also performed using the Vitek^®^ 2 AST-GP80 and AST-GN97 antimicrobial preset card for the characterization of Gram-positive and Gram-negative bacteria, respectively (bioMérieux, Marcy l’Etoile, France), and classified as susceptible (S), intermediate (I) or resistant (R) phenotypes following the manufacturer’s instructions, utilizing phenotypes and MIC distributions obtained from both published literature and internal datasets [21,22]. Fungal identification was performed using a combination of phenotypic microscopical identification, combined with the Vitek^®^ 2 YST fungal identification card.

Descriptive analysis of antimicrobial MDR phenotypes was performed on the most frequent bacterial pathogens detected (*S. pseudintermedius*, *P. aeruginosa, Proteus* (*P.*) *mirabilis*, *E. coli* and *Staphylococcus* (*S.*) *schleiferi*). MDR was defined as the acquired non-susceptibility to at least one antimicrobial in three or more antimicrobial categories [23,24].

### 2.3. Statistical Analysis

A one-sample Kolmogorov–Smirnov test was used to test the normality of the distribution of variables. Continuous variables with normal distribution were presented as mean (standard deviation [SD]; 95% Confidence Interval [CI]); non-normal variables were reported as median (interquartile range [IQR]). Analysis of the association between variables was performed using Kruskal–Wallis, Chi-square or Fisher exact tests. Statistical analysis was performed using IBM^®^ SPSS^®^ Statistic version 26 (IBM Corp., Armonk, NY, USA). *p* value < 0.05 was considered significant.

## 3. Results

### 3.1. Sample Cohort Description

A total of 604 submissions were processed for diagnosis of canine external otitis at the Análisis Veterinarios Eurofins diagnostic laboratory from 2020 to 2022 (2020, *n* = 234; 2021, *n* = 216; 2022, *n* = 154). The age of the dogs ranged from 3 months to 17 years, with a median of 8 years (IQR: 5–11). In regard to age groups, 1.3% of the animals were younger than 1 year, 27.5% from 1 to 5 years and 66.4% older than 5 years. No statistically significant differences between age groups and culture results were observed (*p*-value 0.323). In total, 328 of animals were male (54.3%; 95% CI: 50.3–58.3%) and 276 female (45.7%; 95% CI: 41.7–49.7%). A total of 58 different dog breeds were described. Mixed breed was the most frequently observed breed (*n* = 121; 20.7%; 95% CI: 17.4–24.0%), followed by Labrador Retriever (*n* = 52; 8.9%; 95% CI: 6.6–11.2%), French Bulldog (*n* = 51; 8.7%; 95% CI: 6.4–11.0%), Cocker Spaniel (*n* = 50; 8.6%; 95% CI: 6.3–10.8%), German Shepherd (*n* = 42; 7.2%; 95% CI: 5.1–9.3%) and Yorkshire Terrier (*n* = 39; 6.7%; 95% CI: 4.7–8.7%). Cocker Spaniel was the only breed significantly associated with a higher rate of positive ear culture results (*p*-value 0.05).

### 3.2. Diagnostic Results

Of the samples analyzed, 472 were positive for bacterial or fungal growth (78.1%; 95% CI: 74.8–81.4%). From those positive submissions, a total of 558 microbiological diagnoses were obtained, divided in 421 bacterial (75.4%; 95% CI: 71.8–79.0%) and 137 fungal (24.6%; 95% CI: 20.9–28.1%) unique identifications. 

A total of 39 distinct bacterial and seven distinct fungal species were identified. Table 1 shows the distribution of the detected microorganisms. Within the bacteria examined, *S. pseudintermedius* was the most prevalent microorganism detected (*n* = 128; 22.9%; 95% CI: 19.4–26.4; *p*-value < 0.001), followed by *P. aeruginosa* (*n* = 102; 18.3%; 95% CI: 15.1–21.5; *p*-value < 0.001), *P. mirabilis* (*n* = 48; 8.6%; 95% CI: 6.3–10.9; *p*-value < 0.001), *E. coli* (*n* = 44; 7.9%; 95% CI: 5.6–10.1; *p*-value < 0.001) and *S. schleiferi* (*n* = 16; 2.9%; 95% CI: 1.5–4.3; *p*-value 0.02). The most prevalent fungi and the second most prevalent microorganism observed was *M. pachydermatis* (*n* = 126; 22.6%; 95% CI: 19.1–26.1; *p*-value < 0.001). In addition, *C. albicans* prevalence was also significant within the fungi category (*n* = 4; 0.7%; 95% CI: 0.0–1.4; *p*-value 0.04). 

*S. pseudintermedius* (*p*-value 0.001), *M. pachydermatis* (*p*-value 0.038) and *S. schleiferi* (*p*-value 0.005) were more frequent in animals older than 5 years. No statistically significant differences between sex and microorganism identification were observed.

In total, 46.8% of the bacterial identifications were classified as Gram-positive (*n* = 197; 95% CI 42.0–51.6%), whilst 53.2% were Gram-negative microorganisms (*n* = 224; 95% CI 48.4–58.0%). No statistically significant differences between Gram-positive or negative bacteria were observed.

### 3.3. Antimicrobial Susceptibility, MDR and Coinfection Analysis

The antimicrobial susceptibility profiles of the most frequently identified bacterial pathogens, as detected by the Vitek^®^ 2 system, are detailed in Table 2. The highest frequency of resistant *S. pseudintermedius* isolates was detected against penicillin G (86/125; 68.8%; *p*-value 0.012) and tetracycline (53/127; 41.7%), while the highest susceptibility rates were observed for amoxicillin/clavulanate (34/37; 91.9%; *p*-value 0.005) and nitrofurantoin (124/127; 97.6%; *p*-value < 0.001). A 26.4% resistance rate for oxacillin was also observed for this pathogen (33/125). For *P. aeruginosa*, cefalexin (41/50; 82%: *p*-value 0.001) and ceftiofur (78/95; 82.1; *p*-value < 0.001) displayed the highest rate of resistant phenotypes. The highest susceptibility rates were found for ceftazidime (91/100; 91%; *p*-value < 0.001) and amikacin (94/100; 94%). Doxycycline (44/48; 91.7%; *p*-value < 0.001) and nitrofurantoin (42/48; 87.5%; *p*-value < 0.001) presented the highest susceptibility rates for *P. mirabilis*, while amikacin (43/46; 93.5%) and gentamycin (43/48; 89.6%) were the antimicrobials with the lowest susceptibility rates for this pathogen. For *E. coli*, ampicillin (21/44; 47.7%; *p*-value 0.030) and chloramphenicol (19/44; 43.2%; *p*-value 0.016) showed the highest susceptibility rates, while imipenem (44/44; 100%; *p*-value 0.009) nitrofurantoin (41/44; 93.2%; *p*-value 0.009) displayed the highest susceptibility rates. *S. schleiferi* was only represented by 16 isolates with 100% susceptibility rates in various β-lactams, aminoglycosides, tetracyclines in addition to nitrofurantoin, chloramphenicol and sulfamethoxazole-trimethoprim.

*P. aeruginosa* presented the highest rate of resistant phenotypes (30.7%), followed by *P. mirabilis* (29.4%), *S. pseudintermedius* (25.1%), *E. coli* (19%) and *S. schleiferi* (4.1%). 

When the antimicrobial susceptibility patterns were analyzed by antimicrobial class, tetracyclines presented the highest rate of AMR (38.7% of the isolates), while the aminoglycoside class displayed the highest susceptibility rate (89.2%) (Table 3).

MDR phenotypes were observed in 47% (199/421; 95% CI 42.2–51.8) of the bacterial isolates investigated. *P. mirabilis* presented the highest rate of MDR phenotypes (25/48; 52.1%), followed by *S. pseudintermedius* (66/128; 51.6%), *E. coli* (21/44; 47.7%), *P. aeruginosa* (39/102; 38.2%) and *S. schleiferi* (5/16; 31.3%).

Coinfections were observed in 85 of the positive samples (18.1%; 95% CI: 14.5–21.5), and 32 unique microorganism combinations were observed. The most common coinfection was the combination of *S. pseudintermedius* and *M. pachydermatis* (27/85; 31.8%), followed by *P. aeruginosa* and *M. pachydermatis* (13/85; 15.3%). *M. pachydermatis* was the most common microorganism found in coinfections (61/85; 71.8%; *p*-value < 0.001), followed by *S. pseudintermedius* (39/85; 45.9%; *p*-value < 0.001), *P. aeruginosa* (23/85; 27.1%), *P. mirabilis* (9/85; 10.6%), *E. coli* (7/85; 8.2%) and *S. schleiferi* (2/85; 2.4%).

## 4. Discussion

Canine otitis externa is one of the most prevalent pathologies affecting dogs worldwide. Many microorganisms are involved in the pathogenesis of this condition; therefore, a good knowledge of pathogen distribution in combination with updated antimicrobial susceptibility data is key for an effective management of the disease. For example, in a study performed in 2021 in the United Kingdom, based on the analysis of common pathologies of dogs from a cohort of 22,333 animals, otitis externa ranked second in overall prevalence, right after periodontal disease [25], hence the relevance for a good understanding of the etiological agents involved in this disorder of the ear.

Our study describes the retrospective analysis of diagnostic data obtained from samples submitted to a commercial veterinary diagnostics laboratory of the island of Gran Canaria, Spain from 2020 to 2022. The samples selected in this study represented 2.1% of the total samples processed at the laboratory in the study period selected (604/29,081). Other authors have found relatively similar percentages of ear disease cases when analyzing diagnostic results from large diagnostic facilities. For example, Li et al. [26], observed 1% of samples associated with ear disease in sample set of 20,404 outpatient records. Although the total number of samples submitted for ear disease diagnosis were low overall, the dataset evaluated is representative of the local dog population based on statistical sample size calculation [27]. A 78.1% positive rate (95% CI: 74.8–81.4%) was observed in the population cohort studied. This value is based on submissions with suspect clinical otitis. The percentage on non-diagnosed otitis cases observed could be associated with various factors, including the presence of non-culturable microorganisms, the lack of bacterial or fungal involvement in the cases of otitis and suboptimal sampling, among other reasons. As no clinical records were collected from the sample submission forms, it could also be hypothesized that a proportion of the samples analyzed could have been submitted as a follow-up investigation after treatment, and therefore a lower isolation rate in those cases should be expected. However, similar isolation rates have been described when directly analyzing suspect otitis samples. For example, Tesin et al. [1] observed a detection rate of 88.3% for bacteria and yeast from 60 ear swabs, similarly to Li et al. [26], who found an 86.4% prevalence of otitis externa.

Breed has been described as a highly relevant predisposing factor for the development of otitis externa in dogs [3]. Breeds predisposed to this pathology include Cocker Spaniels, French and English Bulldogs and Labrador Retrievers, among others [28,29,30,31]. In our study, mixed-breed dogs, Labrador Retrievers, French Bulldogs, Yorkshire Terriers, German Shepherds and Cocker Spaniels were the most frequently observed breeds, although only the latter displayed a significant higher rate of otitis externa (*p*-value 0.05).

*S. pseudintermedius* and other coagulase-positive staphylococci, *Malassezia* spp. and *P. aeruginosa*, *Proteus* spp., *E. coli* and beta-haemolytic streptococci are consistently described as the most common microorganisms found in cases of canine otitis externa [1,6,32,33]. Our results are in agreement with the current literature, as *S. pseudintermedius* (22.9%), *M. pachydermatis* (22.6%), *P. aeruginosa* (18.3%), *E. coli* (7.9%) and *S. schleiferi* (2.9%) were the most common microorganisms detected.

Although Gram-negative bacteria were the most commonly identified microorganisms, *Staphylococcus* was the most frequent bacterial genus detected. Staphylococci are common members of the skin microbiota of dogs, predominating in moist areas [34], consistently turning into opportunistic bacterial of otitis externa and other skin infections [35]. Isolation rates of *S. pseudintermedius* in canine otitis externa are variable. Our results are consistent with those observed by Bornand [36], who observed a 23% prevalence in a population of 1118 dogs with otitis. However, other authors describe much higher recovery rates for this pathogen, ranging from 39.2 to 58.8% [1,7,15,37]. *S. schleiferi* (2.9%) and *S. aureus* (1.8%) were also frequently isolated within the *Staphylococcus* genus. *S. schleiferi* is considered as an emerging zoonotic pathogen of humans and animals, with an increasing relevance in canine ear and skin infections [38], while *S. aureus* is responsible for a wide array of pathologies in animals and humans [39]. The prevalence of both pathogens is usually low in canine otitis, ranging from 4.9 to 6.2% [7,40]. However, due to their zoonotic potential and their ability to carry different mechanisms of resistance, attention should be drawn to the role of these two microorganisms in canine otitis externa and public health.

The prevalence of Gram-negative bacteria detected in this study was high, accounting for 40.5% of the total number of microorganisms, and 53.2% of the bacteria detected. This result is consistent with the findings of Terziev et al. [3] and Bugden [32]. However, other authors describe a higher prevalence of Gram-positive microorganisms in cases of canine otitis externa, usually linked to a higher prevalence of *S. pseudintermedius* in their population [1,7,40]. The Gram-negative bacteria identified as part of our study were more diverse, with a total of 13 different genera detected, including various members of the ESKAPE group [41], such as *E. faecium*, *K. pneumoniae*, *A. baumannii*, *P. aeruginosa* and *Enterobacter* spp., which have been previously described in ear and skin infections of dogs [15]. *P. aeruginosa* (18.3%), *P. mirabilis* (8.6%) and *E. coli* (7.9%) were the most common Gram-negative bacteria in our study. The results described agree with other reports defining these microorganisms as the most common Gram-negative bacteria in cases of otitis [7,15,32,40,42]. The prevalence of *P. aeruginosa* described in the present study falls within the detection rates previously described for this pathogen (16.1–35.5%) [32,40,43], while the values for *P. mirabilis* (3.6–6.8%) and *E. coli* (3.2–4.2%) observed in literature for cases of otitis externa [32,40,43] are slightly lower than those observed in this study.

While different reports consider *Corynebacterium* spp. as a relevant microorganism in cases of canine otitis [33,40], no bacteria of this genus were detected in our study. Similarly, other canine otitis prevalence studies did not report this microorganism [32,40]. Interestingly, in a report studying the correlation between cytology, bacterial culture and 16S amplicon profiling for the diagnoses of cases of canine otitis externa [44], the majority of the culture-negative results were diagnosed as *Corynebacterium* spp. based on 16S sequencing, suggesting a more prevalent role of this bacteria in cases of otitis, and the need for adapting isolation protocols to improve *Corynebacterium* spp. detection when using microbiological culture as the technique of choice.

The most prevalent fungi and second most prevalent microorganism observed was *M. pachydermatis* (22.6%). Detection rates for this yeast in literature are broad, ranging from 16% to 67.9% prevalence rates in otitic ears [1,7,20,40,43,45]. *M. pachydermatis* is a common member of the skin microbiota in dogs; however, recent studies focused on understanding the mycobiota of healthy and diseases animals reveal that, in non-affected animals, *Malassezia* is a common although not highly abundant taxa, while in cases of otitis, the mycobiota of the dog ear presents drastic shifts where *M. pachydermatis* becomes overrepresented [46], highlighting its high degree of adaptation to the diseased ear, confirming its central role in canine otitis externa, as observed in our study.

Other fungi detected included members of the *Candida* (*C*.) genus (*C. albicans*, *C. guilliermondii*, *C. parafilopsis*) and *Aspergillus* (*A*.) genus (*A. fumigatus*, *A. niger*), all of them with detection rates lower than 1%. Most of these fungi are considered occasional findings in cases of canine otitis [7,40,47]. To our knowledge, this study represents the first description of *C. guilliermondii* isolated from cases of otitis externa of dogs.

In our study, the rate of infection with multiple pathogens was 18.1%. These results are in accordance with those described by Nocera et al. [15], where the authors observed a 16% rate of coinfections in cases of pyoderma and otitis of dogs. However, other authors have described much higher coinfection rates, ranging from 61.7% [43] to 80% [44]. *M. pachydermatis* (61/85; 71.8%) and *S. pseudintermedius* (39/85; 45.9%) were found in most of the coinfection cases observed, in agreement with previous studies [43]. In spite of the frequency of detection of coinfections, the interaction between pathogens involved in ear infections of dogs and its role in the pathogenesis of canine otitis externa remains extensively uncharacterized [48]. For example, *M. pachydermatis*, apart from other virulence mechanisms, is able to activate transcriptional regulators able to down-regulate the immune response and to modify the function of epidermal cells [49], factors that can potentially facilitate the colonization of the ear skin by other secondary-opportunistic pathogens, a fact that could in part explain the role of this yeast in otitis externa coinfections.

*P. aeruginosa* displayed the highest rate of resistant phenotypes within the five most prevalent bacteria analyzed in our study (30.7%), as observed in previous reports [33]. The highest level of resistance for this bacterial species was observed for β-lactams and tetracyclines, two antimicrobial classes linked to intrinsic resistance in *P. aeruginosa* [50]. Amikacin (94%), ceftazidime (91%), gentamicin (86.3%), marbofloxacin (78%) and polymyxin B (81.4%) were among the antimicrobials with the highest susceptibility rates. These results are in accordance with previous reports [15,51,52]; however, due to the ability of *P. aeruginosa* to develop AMR, adequate use of these antimicrobials is important, as decreased susceptibility to gentamicin and polymyxin B in canine otitis isolates has been already described [43,53].

*P. mirabilis* also presented a high overall rate of resistance in the population analyzed (29.4%). Doxycycline (91.7%), nitrofurantoin (87%) and ampicillin (60%) showed the lowest susceptibility of the antimicrobials tested. A similar level of ampicillin resistance (59%) has also been described for this pathogen in otitis cases [54] and well as in skin infections and abscesses of dogs (50%) [33]. In the same study, performed in Spain, a resistance rate of more than 80% of the isolates to doxycycline was also observed, in agreement with our results. Tetracycline intrinsic resistance has previously described for *P. mirabilis* [55], as well as for nitrofurantoin. The highest susceptibility against this pathogen was observed for aminoglycosides (amikacin: 93.5% and gentamicin: 89.6%), sulfamethoxazole-trimethoprim (83.3%) and selected β-lactams (cefovecin: 80.9% and ceftazidime: 80.4%). Similar results have been observed for gentamicin [40,53]. However, other authors report significantly higher resistance rates for gentamicin (75%) and sulfamethoxazole-trimethoprim (72%), linked in part to the presence of sulfonamide and aminoglycoside specific resistance genes *sul*2 and *aac(6′)-lb-cr*, although the presence of these genes cannot explain by themselves the high degree of resistance observed [54].

*E. coli* presented the lowest resistance rate of the Gram-negative bacteria analyzed (19%). The higher susceptibility of *E. coli*, alone and compared with the main bacteria species observed in our study, agrees with other reports [33]. Ampicillin (47.7%) and chloramphenicol (43.2%) displayed the highest rates of resistance. Resistance to β-lactams has been frequently described in *E. coli* from canine otitis [15,33], linked to the appearance of beta-lactamase-producing strains [56]. Similarly, chloramphenicol resistance in *E. coli* from otitis cases has also been observed [43]. In Spain, chloramphenicol is not used for the treatment of canine otitis; however, florfenicol, another amphenicol-class antibiotic, can be found in topic preparation, so it can be hypothesized that a percentage of these resistant phenotypes could be linked to co-selection of amphenicol-resistant *E. coli*.

*S. pseudintermedius* exhibited the third highest rate of AMR isolates (25.1%). Penicillin G (69%), tetracycline (41.7%) and kanamycin (30.7%) presented the highest resistance phenotype rates. Penicillin G resistance for this opportunist pathogen is a common finding, as well as for tetracycline [1,7,53,57]. Oxacillin resistance was found in 26.4% of the *S. pseudintermedius* isolates, emphasizing the growing concern for the expansion of MRSP strains. Oxacillin (methicillin) resistance in *S. pseudintermedius* obtained from canine otitis cases have been reported worldwide, ranging from 8.9% to 50% [15,40,58,59]. The emergence of this phenotype is of central relevance of animal health, as many MRSP isolates are linked to MDR [60]. In our study, 82.8% of the MRSP isolates were associated with MDR phenotypes, confirming the importance of understanding the epidemiology of this pathogen in order to establish optimal control strategies. In addition, the results from a recent study performed in Brazil confirmed the effective transmission of MRSP among dogs and owners [61], demonstrating the zoonotic risk of this pathogen. In Spain, MRSPs have been isolated from healthy dogs [62], with a 4.6% prevalence. Another study performed in Spain described more than 20% prevalence of oxacillin-resistant *Staphylococcus* spp. from clinical samples, including wound, dermatitis, otitis, abscesses, conjunctivitis and respiratory tract infections [33]; however, the exact percentage of MRSP from otitis cases was not presented. Methicillin-resistant staphylococci in our study also included one *S. schleiferi* (6.25%), three *S. aureus* (30%) and six *S. lentus* (100%), demonstrating the role *Staphylococcus* spp. as a reservoir for methicillin-resistant bacterial phenotypes that could present a risk for animal and public health [63,64].

A high level of bacterial MDR phenotypes were observed in the present study (47%). *P. mirabilis* presented the highest rate of MDR phenotypes (52.1%). This result disagrees with the report presented by Bourély et al. [53], which described *P. mirabilis* as the bacterial pathogen with the lowest MDR phenotype rate (11.8%), with the highest number of isolates susceptible to all antimicrobial tested. On the other hand, *P. aeruginosa* displayed a relatively low MDR phenotype rate in comparison to previously published data (27.1% vs. 66.7–79%) [7,15]. As discussed before, intrinsic AMR has been previously described for these two pathogens and could therefore play a relevant role in the presentation of MDR phenotypes for *P. mirabilis* and *P. aeruginosa*. The frequency of MDR detection for *E. coli* was comparable with current reports [15]. In general terms, the MDR rates described agree with the current AMR scenario, where the increase in prevalence of resistance phenotypes and the public health concern have been the main drivers behind the development of specific national and transnational strategies focused on controlling the alarming distribution of AMR resistant phenotypes [65]. 

This study presents current data on the prevalence and AST of microorganisms isolated from canine otitis externa, providing novel evidence on AMR in the field of canine medicine that could be used for performing informed clinical decisions and treatments, which could benefit both the health of companion animals and humans in an integrated One Health approach.

## 5. Conclusions

Canine otitis externa is one of the most frequent diseases of dogs, with significant welfare and public health concerns. Based on the results discussed, the control of AMR in companion animals must be considered critical, due to the high frequency of resistant phenotypes detected. The widespread detection of MDR and methicillin-resistant bacteria confirms the need for adhering to evidence-based treatments of bacterial and fungal infections. This study adds valuable insights into the complex dynamics of otitis externa in dogs, highlighting the need for ongoing research and evidence-based approaches to address this prevalent condition.

## Figures and Tables

**Table 1 animals-14-00742-t001:** Frequency of distribution of microorganism based on Vitek^®^ 2 system identification.

	*n*	% (95% CI)	*p*-Value		*n*	% (95% CI)	*p*-Value
*Acinetobacter baumannii* *complex*	1	0.2 (−0.2–0.5)	1.00 ^‡^	*Pseudomonas* *fluorescens*	2	0.4 (−0.1–0.9)	1.00 ^‡^
*Aeromonas* *hydrophila-caviae*	1	0.2 (−0.2–0.5)	1.00 ^‡^	*Pseudomonas* *luteola*	1	0.2 (−0.2–0.5)	1.00 ^‡^
*Aspergillus* *fumigatus*	1	0.2 (−0.2–0.5)	0.25 ^‡^	*Pseudomonas oryzihabitans*	1	0.2 (−0.2–0.5)	1.00 ^‡^
*Aspergillus niger*	1	0.2 (−0.2–0.5)	0.25 ^‡^	*Pseudomonas* *stutzeri*	1	0.2 (−0.2–0.5)	1.00 ^‡^
*Burkholderia* *cepacia*	2	0.4 (−0.1–0.9)	1.00 ^‡^	*Serratia fonticola*	1	0.2 (−0.2–0.5)	1.00 ^‡^
*Candida albicans*	4	0.7 (0.0–1.4)	**0.04 ^‡^**	*Serratia* *liquefaciens*	1	0.2 (−0.2–0.5)	1.00 ^‡^
*Candida* *guilliermondii*	1	0.2 (−0.2–0.5)	0.25 ^‡^	*Serratia* *marcescens*	2	0.4 (−0.1–0.9)	1.00 ^‡^
*Candida* *parafilopsis*	1	0.2 (−0.2–0.5)	0.25 ^‡^	*Sphingomonas paucimobilis*	1	0.2 (−0.2–0.5)	1.00 ^‡^
*Candida* spp.	2	0.4 (−0.1–0.9)	0.06 ^‡^	*Staphylococcus* *aureus*	10	1.8 (0.7–2.9)	0.13 ^‡^
*Citrobacter* *freundii*	1	0.2 (−0.2–0.5)	1.00 ^‡^	*Staphylococcus chromogenes*	1	0.2 (−0.2–0.5)	1.00 ^‡^
*Citrobacter koseri*	4	0.7 (0.0–1.4)	0.58 ^‡^	*Staphylococcus* coag Neg	2	0.4 (−0.1–0.9)	1.00 ^‡^
*Enterobacter* *cloacae*	3	0.5 (−0.1–1.1)	1.00 ^‡^	*Staphylococcus epidermidis*	4	0.7 (0.0–1.4)	0.58 ^‡^
*Enterococcus* *faecalis*	11	1.9 (0.8–3.1)	0.07 ^‡^	*Staphylococcus haemolyticus*	5	0.9 (0.1–1.7)	0.34 ^‡^
*Enterococcus* *faecium*	1	0.2 (−0.2–0.5)	1.00 ^‡^	*Staphylococcus hominis* ssp. *hominis*	2	0.4 (−0.1–0.9)	1.00 ^‡^
*Enterococcus* spp.	1	0.2 (−0.2–0.5)	1.00 ^‡^	*Staphylococcus intermedius*	3	0.5 (−0.1–1.1)	1.00 ^‡^
*Escherichia coli*	44	7.9 (5.6–10.1)	**<0.001 ^+^**	*Staphylococcus lentus*	6	1.1 (0.2–1.9)	0.34 ^‡^
*Haemophilus* *haemolyticus*	1	0.2 (−0.2–0.5)	1.00 ^‡^	*Staphylococcus pseudintermedius*	128	22.9 (19.4–26.4)	**<0.001 ^+^**
*Haemophilus parainfluenza*	1	0.2 (−0.2–0.5)	1.00 ^‡^	*Staphylococcus saprophyticcus*	1	0.2 (−0.2–0.5)	1.00 ^‡^
*Klebsiella* *pneumoniae*	4	0.7 (0.0–1.4)	0.58 ^‡^	*Staphylococcus schleiferi*	16	2.9 (1.5–4.3)	**0.02** ^‡^
*Malassezia furfur*	1	0.2 (−0.2–0.5)	0.25 ^‡^	*Staphylococcus warneri*	1	0.2 (−0.2–0.5)	1.00 ^‡^
*Malassezia* *pachydermatis*	126	22.6 (19.1–26.1)	**<0.001 ^+^**	*Staphylococcus* *xylosus*	2	0.4 (−0.1–0.9)	1.00 ^‡^
*Proteus mirabilis*	48	8.6 (6.3–10.9)	**<0.001 ^+^**	*Streptococcus* *canis*	1	0.2 (−0.2–0.5)	1.00 ^‡^
*Providencia* *stuartii*	2	0.4 (−0.1–0.9)	1.00 ^‡^	*Streptococcus* *mutans*	1	0.2 (−0.2–0.5)	1.00 ^‡^
*Pseudomonas* *aeruginosa*	102	18.3 (15.1–21.5)	**<0.001 ^+^**	*Streptococcus* *parasanguinis*	1	0.2 (−0.2–0.5)	1.00 ^‡^

^‡^ Fisher exact test; **^+^** Chi-squared test.

**Table 2 animals-14-00742-t002:** Distribution of antimicrobial susceptibility profiles among the most prevalent bacterial pathogens detected using Vitek^®^ 2 system. Phenotypes are distributed as susceptible (S), intermediate (I) or resistant (R).

	*S. pseudintermedius*			*P. aeruginosa*			*P. mirabilis*			*E. coli*			*S. schleiferi*		
	S	I	R	S	I	R	S	I	R	S	I	R	S	I	R
**CL**				18% (*n* = 9)		82% (*n* = 41)	41.3% (*n* = 19)	19.6% (*n* = 9)	39.1% (*n* = 18)	63.6% (*n* = 28)	4.5% (*n* = 2)	31.8% (*n* = 14)			
**CVN**	83.6% (*n* = 107)	3.9% (*n* = 5)	12.5% (*n* = 16)	27.8% (*n* = 15)	5.6% (*n* = 3)	66.7% (*n* = 36)	80.9% (*n* = 38)	6.4% (*n* = 3)	12.8% (*n* = 6)	72.7% (*n* = 32)	2.3% (*n* = 1)	25% (*n* = 11)	100% (*n* = 16)		
**CAZ**				91% (*n* = 91)	1% (*n* = 1)	8% (*n* = 8)	80.4% (*n* = 37)	13% (*n* = 6)	6.5% (*n* = 3)	83.7% (*n* = 36)	2.3% (*n* = 1)	14% (*n* = 6)			
**CPD**				42.9% (*n* = 12)		57.1% (*n* = 16)	78.3% (*n* = 36)		21.7% (*n* = 10)	83.7% (*n* = 36)		16.3% (*n* = 7)			
**CTF**	87.5% (*n* = 7)		12.5% (*n* = 1)	11.6% (*n* = 11)	6.3% (*n* = 6)	82.1% (*n* = 78)	68.2% (*n* = 30)	18.2% (*n* = 8)	13.6% (*n* = 6)	70% (*n* = 33)	9.1% (*n* = 4)	15.9% (*n* = 7)	100% (*n* = 5)		
**AMC**	91.9% (*n* = 34)		8.1% (*n* = 3)	37.5% (*n* = 9)		62.5% (*n* = 15)	64.6% (*n* = 31)	6.3% (*n* = 3)	29.2% (*n* = 14)	90.9% (*n* = 40)	2.3% (*n* = 1)	6.8% (*n* = 3)			
**AMP**				29.2% (*n* = 7)		70.8% (*n* = 17)	40% (*n* = 18)		60% (*n* = 27)	52.3% (*n* = 23)		47.7% (*n* = 21)			
**P**	31% (*n* = 39)		69% (*n* = 87)										100% (*n* = 16)		
**OXA**	73.6% (*n* = 92)		26.4% (*n* = 33)										93.8% (*n* = 15)		6.3% (*n* = 1)
**IPM**				84% (*n* = 84)	4% (*n* = 4)	12% (*n* = 12)	21.7% (*n* = 10)	58.7% (*n* = 27)	19.6% (*n* = 9)	100% (*n* = 44)					
**AMI**				94% (*n* = 94)		6% (*n* = 6)	93.5% (*n* = 43)		6.5% (*n* = 3)	90.9% (*n* = 40)	2.3% (*n* = 1)	6.8% (*n* = 3)			
**GEN**	80% (*n* = 100)	2.4% (*n* = 3)	17.6% (*n* = 22)	86.3% (*n* = 88)	1% (*n* = 1)	12.7% (*n* = 13)	89.6% (*n* = 43)	4.2% (*n* = 2)	6.3% (*n* = 3)	88.6% (*n* = 39)		11.4% (*n* = 5)	100% (*n* = 16)		
**K**	68.5% (*n* = 87)	0.8% (*n* = 1)	30.7% (*n* = 39)										100% (*n* = 16)		
**N**	75.8% (*n* = 97)	6.3% (*n* = 8)	18% (*n* = 23)										100% (*n* = 16)		
**CIP**				70.3% (*n* = 71)	2% (*n* = 2)	27.7% (*n* = 28)	76.1% (*n* = 35)	2.2% (*n* = 1)	21.7% (*n* = 10)	77.3% (*n* = 34)	6.8% (*n* = 3)	15.9% (*n* = 7)			
**ENR**	65.4% (*n* = 83)	12.6% (*n* = 16)	22% (*n* = 28)	34.7% (*n* = 35)	44.6% (*n* = 45)	20.8% (*n* = 21)	72.9% (*n* = 35)	4.2% (*n* = 2)	22.9% (*n* = 11)	81.8% (*n* = 36)	9.1% (*n* = 4)	9.1% (*n* = 4)	56.3% (*n* = 9)	18.8% (*n* = 3)	25% (*n* = 4)
**MAR**	73.4% (*n* = 94)	7.8% (*n* = 10)	18.8% (*n* = 24)	78% (*n* = 78)	11% (*n* = 11)	11% (*n* = 11)	77.1% (*n* = 37)	16.7% (*n* = 8)	6.3% (*n* = 3)	84.1% (*n* = 37)	4.5% (*n* = 2)	11.4% (*n* = 5)	68.8% (*n* = 11)	6.3% (*n* = 1)	25% (*n* = 4)
**ERY**	65.6% (*n* = 84)	4.7% (*n* = 6)	29.7% (*n* = 38)										100% (*n* = 16)		
**CLI**	67.5% (*n* = 85)	3.2% (*n* = 4)	29.4% (*n* = 37)										93.8% (*n* = 15)		6.3% (*n* = 1)
**DOX**	59.8% (*n* = 76)	12.6% (*n* = 16)	27.6% (*n* = 35)	55.6% (*n* = 20)		44.4% (*n* = 16)	6.3% (*n* = 3)	2.1% (*n* = 1)	91.7% (*n* = 44)	70.5% (*n* = 31)	6.8% (*n* = 3)	22.7% (*n* = 10)	100% (*n* = 16)		
**TET**	57.5% (*n* = 73)	0.8% (*n* = 1)	41.7% (*n* = 53)										100% (*n* = 16)		
**NIT**	97.6% (*n* = 124)		2.4% (*n* = 3)	56.3% (*n* = 18)		43.8% (*n* = 14)	10.4% (*n* = 5)	2.1% (*n* = 1)	87.5% (*n* = 42)	93.2% (*n* = 41)		6.8% (*n* = 3)	100% (*n* = 16)		
**CHL**	78.9% (*n* = 101)	1.6% (*n* = 2)	19.5% (*n* = 25)	59.4% (*n* = 19)	3.1% (*n* = 1)	37.5% (*n* = 12)	60.4% (*n* = 29)	2.1% (*n* = 1)	37.5% (*n* = 18)	50% (*n* = 22)	6.8% (*n* = 3)	43.2% (*n* = 19)	100% (*n* = 16)		
**SXT**	82% (*n* = 105)		18% (*n* = 23)	70% (*n* = 21)	6.7% (*n* = 2)	23.3% (*n* = 7)	83.3% (*n* = 40)		16.7% (*n* = 8)	76.7% (*n* = 33)		23.3% (*n* = 10)	100% (*n* = 16)		
**PB**				81.4% (*n* = 70)		18.6% (*n* = 16)	83.3% (*n* = 5)		16.7% (*n* = 1)	73.8% (*n* = 31)	7.1% (*n* = 3)	19% (*n* = 8)			
**Total SRI**	71.2%	3.7%	25.1%	62.9%	6.4%	30.7%	61.6%	9%	29.4%	77.3%	3.7%	19%	94.3%	1.6%	4.1%

Antimicrobial abbreviations: CL = Cefalexin; CVN = Cefovecin; CAZ = Ceftazidime; CPD = Cefpodoxime; CTF = Ceftiofur; AMC = Amoxicillin/clavulanate; AMP = Ampicillin; P = Penicillin G; OXA = Oxacillin; IPM = Imipenem; AMI = Amikacin; GEN = Gentamicin; K = Kanamycin; N = Neomycin; CIP = Ciprofloxacin; ENR = Enrofloxacin; MAR = Marbofloxacin; ERY = Erythromycin; CLI = Clindamycin; DOX = Doxycycline; TET = Tetracycline; NIT = Nitrofurantoin; CHL = Chloramphenicol; SXT = Sulfamethoxazole-trimethoprim; PB = Polymyxin B.

**Table 3 animals-14-00742-t003:** Antimicrobial susceptibility phenotypes per antimicrobial class.

	*S. pseudintermedius*		*P. aeruginosa*		*P. mirabilis*		*E. coli*		*S. schleiferi*		Mean	
	S (%)	R (%)	S (%)	R (%)	S (%)	R (%)	S (%)	R (%)	S (%)	R (%)	S (%)	R (%)
**β-Lactams**	73.5	25.7	32.5	53.7	59.4	25.3	77.1	22.5	98.5	6.3	68.2	26.7
**Aminoglycosides**	74.8	22.1	90.2	9.4	91.6	6.4	89.8	9.1	100	-	89.2	9.4
**Fluroquinolones**	69.4	20.4	61	19.8	75.4	17	81.1	39.4	62.6	25	69.9	24.3
**MLS ***	66.6	39.8							96.9	6.3	81.7	23
**Tetracyclines**	58.7	34.7	55.6	44.4	6.3	91.7	70.5	22.7	100	-	58.2	38.7
**Nitrofurans**	97.6	2.4	56.3	43.8	10.4	87.5	93.2	6.8	100	-	71.5	28.1
**Amphenicols**	78.9	19.5	59.4	37.5	60.4	37.5	50	43.2	100	-	69.7	27.5
**SXT**	0.8	18	70	23.3	83.3	16.7	76.7	23.3	100	-	66.2	16.3
**Polypeptides**			81.4	18.6	83.3	16.7	73.8	19			79.5	18.1

S: Susceptible; R: Resistant. SXT = Sulfamethoxazole-trimethoprim. MLS: macrolide-lincosamide-streptogramin. * Classified as a single class due to the similar resistance mechanisms.

## Data Availability

The data presented in this study are available on request from the corresponding author.

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
