# Peer review of "Microbiological Survey and Evaluation of Antimicrobial Susceptibility Patterns of Microorganisms Obtained from Suspect Cases of Canine Otitis Externa in Gran Canaria, Spain"

_animals, 2024, doi:10.3390/ani14050742_

Round 1
Reviewer 1 Report
Comments and Suggestions for Authors
The authors have performed a microbiological survey from the clinical cases of canine otitis externa from the year 2020 to 2022. The authors have also described and categorized sensitivity and resistance to multiple antibiotics. The phenotypes of susceptibility profile has been divided in three categories, sensitive, intermediate to being resistant, and resistant. Depending on the survival rate performing in vitro assays.
However, the authors have not used any positive control representing microflora of a healthy external ear canal of cats or dogs. The authors are encouraged to discuss this from previous publications highlighting the differences and adding value for readers understanding.
Additionally, the authors are requested to appropriately formate the tables 1,2, and 3 to have all columns fit in the paper for the readers. The authors are encouraged to create heat maps to represent the data in a visually more represented format.
Author Response
REVIEWER 1
The authors have performed a microbiological survey from the clinical cases of canine otitis externa from the year 2020 to 2022. The authors have also described and categorized sensitivity and resistance to multiple antibiotics. The phenotypes of susceptibility profile has been divided in three categories, sensitive, intermediate to being resistant, and resistant. Depending on the survival rate performing in vitro assays.
However, the authors have not used any positive control representing microflora of a healthy external ear canal of cats or dogs. The authors are encouraged to discuss this from previous publications highlighting the differences and adding value for readers understanding.
Thanks a lot for the valuable comment. The implication of commensal microorganisms in cases of canine otitis externa has been addressed in lines 64 to 71. Positive results are based on the criteria of expert microbiologist and the results of automated bacterial ID systems. Samples selected included samples submitted for suspect cases of otitis externa, however, even if the detection rates of different microorganism are described, we never establish as part of our discussion any link between microbiological results and the development of otitis. We do not describe any clinical aspect of the disease as part of the data analysis presented, while the role of commensality is described, so the reader can understand the relevance of the data presented.
Additionally, the authors are requested to appropriately formate the tables 1,2, and 3 to have all columns fit in the paper for the readers. The authors are encouraged to create heat maps to represent the data in a visually more represented format.
Thanks for the comment. We have tried to adjust the tables as requested, taking into consideration the high amount of data presented. Regarding the heat maps, we agree with the comment from the reviewer, however, we decided to add percentages and confidence intervals as a representation of our results so they can be easily compared with the available literature.
Reviewer 2 Report
Comments and Suggestions for Authors
This ms reports a local microbiological investigation (retrospective) of organisms associated with otitis externa in dogs. The findings are helpful to understand the microbial etiology and antimicrobial resistance. Some clarities can be helpful for improving the ms. Major and editorial comments are given below:
L33. Abstract. Please include information on the total number of isolates assessed to understand the importance/scale of the study.
L66/L67/L73/L77/L95/L332/L333. No need to write in brackets the uppercase letter (E, M. C, P, S, A) of the microorganisms per scientific writing in the microbiology field. Please delete them. Full spelling in their first appearance (first “Escherichia coli”; than “E. coli”).
L116. To understand the significance of the study finding, please add information on “a veterinary clinical diagnostics laboratory (Animal Lab S.L.)” for its scale/serving region/client size.
L122. The unit of years is need for those numeric values (<1. 1to 5, >5).
L133/L215. Write “susceptible (S)” for “sensitive (S)”. Understand English/French/Spanish difference for the translation. Also, change in Foodnote 1 of table 3.
L134. “manufacturer’s instructions”: please provide which standards are used, e.g., CLSI or EUCAST (but do full spell CLSI/EUCAST when providing).
L136. MDR is defined by using a key reference in the field (human medicine). However, the reference does not cover two pathogens, S. pseudintermedius and S. schleiferi. Mover, P. mirabilis is know for multiple intrinsic resistance. Whether intrinsic resistance is excluded in grouping MDR needs to be described. Please provide additional information for clarity. The authors may also wish refer to “i) Applying definitions for multidrug resistance, extensive drug resistance and pandrug resistance to clinically significant livestock and companion animal bacterial pathogens. J Antimicrob Chemother. 2018;73(6):1460-1463. doi: 10.1093/jac/dky043. ii) Antimicrobial susceptibility testing in veterinary medicine: performance, interpretation of results, best practices and pitfalls. One Health Advances 2023;1:26. Doi:10.1186/s44280-023-00024-w).
L223. Write “AMR” for “antimicrobial resistance” (introduced in L88).
L225. Write “susceptibility” for “sensitivity”. Please check the ms throughout.
L277/L352/L369/L380/Table 3: Write Greek “β” for “beta” (if so, Beta-lactams would not be listed as the first in Table 3; write “β-Lactams [L as uppercase letter] in Table 3 after having Greek β).
L420. No need to write “One Health” in italic.
References – Italicise microorganisms and use lower case letters for “Pseudintermedius” (Ref 17), “Spp”.) (Ref 18, 19), “Schleiferi” (ref 34), “Aeruginosa “ (Ref 48), “Mirabilis “ (Ref 50), etc. Ref 14 has incorrect title (“&”). Please check carefully. The journal editor may have incorrectly formatted the references.
Comments on the Quality of English LanguageEnglish expression in terms of scientific writing can be improved. See some editorial comments to authors above.
Author Response
REVIEWER 2
This ms reports a local microbiological investigation (retrospective) of organisms associated with otitis externa in dogs. The findings are helpful to understand the microbial etiology and antimicrobial resistance. Some clarities can be helpful for improving the ms. Major and editorial comments are given below:
L33. Abstract. Please include information on the total number of isolates assessed to understand the importance/scale of the study.
DONE
L66/L67/L73/L77/L95/L332/L333. No need to write in brackets the uppercase letter (E, M. C, P, S, A) of the microorganisms per scientific writing in the microbiology field. Please delete them. Full spelling in their first appearance (first “Escherichia coli”; than “E. coli”).
DONE
L116. To understand the significance of the study finding, please add information on “a veterinary clinical diagnostics laboratory (Animal Lab S.L.)” for its scale/serving region/client size.
DONE
L122. The unit of years is need for those numeric values (<1. 1to 5, >5).
DONE
L133/L215. Write “susceptible (S)” for “sensitive (S)”. Understand English/French/Spanish difference for the translation. Also, change in Foodnote 1 of table 3.
DONE
L134. “manufacturer’s instructions”: please provide which standards are used, e.g., CLSI or EUCAST (but do full spell CLSI/EUCAST when providing).
Modified as suggested
L136. MDR is defined by using a key reference in the field (human medicine). However, the reference does not cover two pathogens, S. pseudintermedius and S. schleiferi. Mover, P. mirabilis is know for multiple intrinsic resistance. Whether intrinsic resistance is excluded in grouping MDR needs to be described. Please provide additional information for clarity. The authors may also wish refer to “i) Applying definitions for multidrug resistance, extensive drug resistance and pandrug resistance to clinically significant livestock and companion animal bacterial pathogens. J Antimicrob Chemother. 2018;73(6):1460-1463. doi: 10.1093/jac/dky043. ii) Antimicrobial susceptibility testing in veterinary medicine: performance, interpretation of results, best practices and pitfalls. One Health Advances 2023;1:26. Doi:10.1186/s44280-023-00024-w).
Thanks for the comment. The reference has been added. The presence of intrinsic resistance for both P. aeruginosa and P. mirabilis is already discussed in the manuscript (L360-385). The intrinsically resistant phenotypes were not excluded from the analysis, as not 100% of the isolates were resistant in all cases to those antimicrobials, hence the need for the inclusion in the MDR group, coupled with the explanation presented in the discussion section, where the role of intrinsic resistance is explained for both pathogens.
L223. Write “AMR” for “antimicrobial resistance” (introduced in L88).
DONE
L225. Write “susceptibility” for “sensitivity”. Please check the ms throughout.
DONE
L277/L352/L369/L380/Table 3: Write Greek “β” for “beta” (if so, Beta-lactams would not be listed as the first in Table 3; write “β-Lactams [L as uppercase letter] in Table 3 after having Greek β).
DONE
L420. No need to write “One Health” in italic.
DONE
References – Italicise microorganisms and use lower case letters for “Pseudintermedius” (Ref 17), “Spp”.) (Ref 18, 19), “Schleiferi” (ref 34), “Aeruginosa “ (Ref 48), “Mirabilis “ (Ref 50), etc. Ref 14 has incorrect title (“&”). Please check carefully. The journal editor may have incorrectly formatted the references.
DONE
Reviewer 3 Report
Comments and Suggestions for Authors
The manuscript "Microbiological survey and evaluation of antimicrobial susceptibility patterns of microorganisms obtained from suspect cases of canine otitis externa in Gran Canaria, Spain" focus an important and common subject related to companion animal’s veterinary practice, and could help practitioners how to best manage canine otitis cases. It also contributes to an active surveillance regarding antimicrobial resistance, so crucial to the “one Health” issue.
The Simple Summary and Abstract are correct and elucidates the content of the manuscript.
Line 20 – correct “… treatment” to treated
The introduction section is satisfactory.
The materials and methods seem adequate.
The results are sound.
Lines 197-198 – “For P. aeruginosa, cefalexin (41/50; 82%: p-value 0.001) and ceftiofur (78/95; 82.1; p-value <0.001).” The sentence seems incomplete…
The discussion is correct.
Lines 290-291 – “S. schleiferi in considered” correct to: S. schleiferi is considered
Author Response
REVIEWER 3
The manuscript "Microbiological survey and evaluation of antimicrobial susceptibility patterns of microorganisms obtained from suspect cases of canine otitis externa in Gran Canaria, Spain" focus an important and common subject related to companion animal’s veterinary practice, and could help practitioners how to best manage canine otitis cases. It also contributes to an active surveillance regarding antimicrobial resistance, so crucial to the “one Health” issue.
The Simple Summary and Abstract are correct and elucidates the content of the manuscript.
Line 20 – correct “… treatment” to treated
DONE
The introduction section is satisfactory.
The materials and methods seem adequate.
The results are sound.
Lines 197-198 – “For P. aeruginosa, cefalexin (41/50; 82%: p-value 0.001) and ceftiofur (78/95; 82.1; p-value <0.001).” The sentence seems incomplete…
DONE
The discussion is correct.
Lines 290-291 – “S. schleiferi in considered” correct to: S. schleiferi is considered
DONE